# SSL5-AnxA5 Fusion Protein Constructed Based on Human Atherosclerotic Plaque scRNA-Seq Data Preventing the Binding of Apoptotic Endothelial Cells, Platelets, and Inflammatory Cells

**DOI:** 10.3390/biomedicines13010008

**Published:** 2024-12-24

**Authors:** Yifei Zhao, Xingyu He, Teng Hu, Tianli Xia, Fangyang Huang, Changming Li, Yiming Li, Fei Chen, Mao Chen, Jun Ma, Yong Peng

**Affiliations:** 1Department of Cardiology, West China Hospital, Sichuan University, Chengdu 610041, China; 2Clinical Medical College, Chengdu University, Chengdu 610106, China

**Keywords:** stable angina, acute coronary syndrome, fusion protein, P-selectin, superantigen-like 5

## Abstract

**Background and aims:** Coronary obstruction following plaque rupture is a critical pathophysiological change in the progression of stable angina (SAP) to acute coronary syndrome (ACS). The accumulation of platelets and various inflammatory cells on apoptotic endothelial cells is a key factor in arterial obstruction after plaque rupture. Through single-cell sequencing analysis (scRNA-seq) of plaques from SAP and ACS patients, we identified significant changes in the annexin V and P-selectin glycoprotein ligand 1 pathways. Staphylococcal superantigen-like 5 (SSL5) is an optimal antagonist P-selectin glycoprotein ligand 1 (PSGL1), while annexin V (AnxA5) can precisely detect dead cells in vivo. We constructed the SSL5-AnxA5 fusion protein and observed its role in preventing the interaction between apoptotic endothelial cells, platelets, and inflammatory cells. **Methods:** The scRNA-seq data were extracted from the Gene Expression Omnibus (GEO) database. Single-cell transcriptome analysis results and cell–cell communication were analyzed to identify the ACS and SAP cell clusters and elucidate the intercellular communication differences. Then, we constructed and verified a fusion protein comprising SSL5 and AnxA5 domains via polymerase chain reaction (PCR) and Western blot. The binding capacity of the fusion protein to P-selectin and apoptotic cells was evaluated by flow cytometry and AnxA5-FITC apoptosis detection kit, respectively. Furthermore, co-incubation and immunofluorescence allowed us to describe the mediation effect of it between inflammatory cells and endothelial cells or activated platelets. **Results:** Our analysis of the scRNA-seq data showed that *SELPLG* (PSGL1 gene) and *ANNEXIN* had higher information flowing in ACS compared to SAP. The *SELPLG* signaling pathway network demonstrated a higher number of interactions in ACS, while the *ANNEXIN* signaling pathway network revealed stronger signaling from macrophages toward monocytes in ACS compared to SAP. Competition binding experiments with P-selectin showed that SSL5-AnxA5 induced a decrease in the affinity of PSGL1. SSL5-AnxA5 effectively inhibited the combination of endothelial cells with inflammatory cells and the interaction of activated platelets with inflammatory cells. Additionally, this fusion protein exhibited remarkable capability in binding to apoptotic cells. **Conclusions:** The bifunctional protein SSL5-AnxA5 exhibits promising potential as a protective agent against local inflammation in arterial tissues, making it an excellent candidate for PSGL1-related therapeutic interventions.

## 1. Introduction

Cardiovascular diseases (CVDs) have emerged as a formidable global health challenge [1]. Among the range of CVDs, coronary heart disease (CHD) occupies a significant part and is traditionally categorized into stable angina pectoris (SAP) and acute coronary syndrome (ACS), as evident [2,3]. SAP is caused by myocardial ischemia due to plaque formation, while ACS results from myocardial infarction triggered by plaque rupture and obstruction, which is significantly related to the interaction of apoptotic endothelial cells with inflammatory cells [4,5]. In order to investigate potential therapeutic strategies at the molecular level for patients from SAP to ACS, we conducted an in-depth analysis of single-cell sequencing (scRNA-seq) data from plaque samples of SAP and ACS patients in the Gene Expression Omnibus (GEO) database. Our findings revealed upregulation of the *SELPLG* and *ANNEXIN* signaling pathway in both the SAP and ACS populations, accompanied by conjecture about whether there exists specific proteins related to these pathways that contribute to the therapy of CHD.

P-selectin, which is also known as CD62P, is expressed and serves as a key mediator in promoting the aggregation of endothelial cells, platelets, and leukocytes [6,7,8,9]. Its natural ligand, P-selectin glycoprotein ligand-1 (PSGL1), is a protein encoded by the *SELPLG* gene. Notably, staphylococcal superantigen-like protein-5 (SSL5) has recently been identified as an effective antagonist of PSGL1. It can specifically bind to PSGL1, competitively antagonizing the interaction of PSGL1 and selectin, thereby significantly inhibiting leukocyte adhesion without inducing the non-specific immune system activation associated with standard superantigens [10,11]. SSL5 holds great promise as a therapeutic agent for preventing endothelial cell, platelet, and leukocyte aggregation. However, addressing the challenge of achieving effective local drug concentrations is crucial to ensure its clinical efficacy.

In the *ANNEXIN* signaling pathway, a calcium-dependent phospholipid-binding protein called AnxA5 caught our eye. Previous studies have illustrated that AnxA5 exhibited specific binding affinity toward phosphatidylserine exposed on the membrane of apoptotic cells [12,13,14,15]. Given its ability to selectively recognize and bind apoptotic cells within damaged areas, AnxA5 represents a promising targeting molecule for apoptotic endothelial cells in ACS. Thus, we o a hypothesis that through construction of the fusion protein SSL5-AnxA5, the SSL5 will aggregate in the injured endothelial cells to elevate the local drug concentration and achieve the best therapeutic effect, which is mediated by the specific location ability of AnxA5.

In this study, based on our analysis of the single-cell sequencing database for ACS and SAP patients, we engineered a fusion protein, SSL5-AnxA5, preventing the interaction between apoptotic endothelial cells and platelets, as well as inflammatory cells, during plaque rupture. Our research may provide new therapeutic strategies and insights for delaying the progression of SAP to ACS.

## 2. Materials and Methods

### 2.1. Public Data Collection

Raw single-cell RNA sequencing (scRNA-seq) data were obtained from the Gene Expression Omnibus (GEO) database (https://www.ncbi.nlm.nih.gov/geo/query/acc.cgi (accessed on 15 October 2024)) under accession number GSE184073. This dataset comprises scRNA-seq data from cells extracted from atherosclerotic plaques of the coronary arteries of patients with SAP and ACS.

### 2.2. Single-Cell Transcriptome Analysis

The raw expression data for each sample were processed using the Seurat package [16]. Cells with fewer than 200 detected genes were excluded to remove low-quality cells. Additionally, cells with a high proportion of mitochondrial transcripts (>25%) were filtered out, as elevated mitochondrial gene expression is typically associated with cell stress or apoptosis. The expression matrix of the remaining cells was normalized using the “LogNormalize” method with a scale factor of 10,000 [17]. Batch effects were corrected using the Harmony algorithm, and dimensionality reduction was performed via principal component analysis (PCA) [18], and the top 30 principal components were retained for downstream analysis. Uniform manifold approximation and projection (UMAP) was subsequently applied for visualization of the cell populations in a two-dimensional space. Cell clustering was performed using the Louvain algorithm, which identified distinct cell populations based on the first 30 principal components. The resolution for the clustering was automatically optimized based on the dataset to ensure biologically meaningful clusters. The final dataset was found to contain 16 distinct clusters. These clusters were visualized using UMAP plots, and the cells were further grouped by sample type to compare the population distributions between conditions. The cell clusters were annotated using marker genes identified with the FindAllMarkers function (min.pct = 0.25, logfc.threshold = 0.25). Automated cell type prediction was performed using the SingleR package(version 1.1.3), with the Human Primary Cell Atlas dataset, obtained via the HumanPrimaryCellAtlasData function from the celldex package(version 1.1.3) [19]. The most frequently predicted cell types were assigned to each Seurat cluster, including annotations such as “Monocytes”.

### 2.3. Cell–Cell Communication Analysis

Cell–cell communication analysis was conducted using the CellChat R package (version 1.1.3) [20]. Single-cell RNA-seq data were divided into two experimental groups, ACS and SAP. Communication networks were constructed by creating separate CellChat objects for each group using their respective cell type annotations. The CellChatDB human database was used to infer the interactions, and overexpressed ligands and receptors were identified within each group.

For each dataset, the probabilities of cell–cell interactions were computed using the computeCommunProb function, with the gene expression thresholds adjusted to trim low-expression genes. Interactions involving fewer than 10 cells were filtered out (min.cells = 10). The pathway communication probabilities were then calculated with the computeCommunProbPathway function, followed by aggregation of the communication networks using aggregateNet. The centrality measures for each cell type were computed with netAnalysis_computeCentrality to assess their role as signaling sources or receivers.

### 2.4. Preparation of SSL5-AnxA5

The amino acid sequence was determined, and a HisX6 tag was added. The SSL5 gene, linker and AnxA5 genes were linked by overlapping polymerase chain reaction (PCR). All the linked sequences were cloned into a pet28a plasmid. One microliter of the recombinant pet28a vector was used to transform Rosetta (DE3). Then, a single bacterial colony expressing Rosetta (DE3) was selected by the plate screening method and used for culture and expression. Appendix A presents the following steps of the preparation of SSL5-AnxA5. We performed SDS-PAGE detection after the cells with the most efficient fusion protein expression and induction times were selected. The gel revealed bands representing the generated fusion protein. The preliminary purification was effective, but degradation bands remained. To solve this problem, we centrifuged the bacteria and optimized the purification conditions after bacterial breakage. The SDS-PAGE gel revealed the target protein. The selected colony was used to expand the expression. The process of purification was presented as follows and Appendix A describes the specific composition of buffers. Equilibration: Take 5 mL of Ni-NTA. Equilibrate the column by washing with 5 column volumes of Binding Buffer at a flow rate of 5 mL/min. Loading: After incubating the material with the sample for 1 h, load the sample onto the column and collect the flow-through. Column washing: Wash the column with Binding Buffer at a flow rate of 5 mL/min. Impurity removal: Eliminate nonspecific binding by washing the column with Washing Buffer and collect the flow-through. Elution: Elute the target protein using Elution Buffer and collect the eluate. Purification analysis: Process the crude protein flow-through from the washing steps and the eluate separately, prepare the samples, and analyze by SDS-PAGE. After that, Western blotting was performed. The target protein was confirmed with a His-labelled primary antibody. Further tests showed that the concentration of the protein was 1.77 mg/mL. This method has been used in previous work [21]. The process for preparation of SSL5-AnxA5 was completed by Shanghai Sangon Biotech Company (order serial number:25026830(PR018896)).

The amino acid sequence of the SSL5-AnxA5: protein length = 560, MW = 63 kDa, predicted pl = 8.26, MGSSHHHHHHSSGLVPRGSHMMAQVLRGTVTDFPGFDERADAETLRKAMKGLGTDEESILTLLTSRSNAQRQEISAAFKTLFGRDLLDDLKSELTGKFEKLIVALMKPSRLYDAYELKHALKGAGTNEKVLTEIIASRTPEELRAIKQVYEEEYGSSLEDDVVGDTSGYYQRMLVVLLQANRDPDAGIDEAQVEQDAQALFQAGELKWGTDEEKFITIFGTRSVSHLRKVFDKYMTISGFQIEETIDRETSGNLEQLLLAVVKSIRSIPAYLAETLYYAMKGAGTDDHTLIRVMVSRSEIDLFNIRKEFRKNFATSLYSMIKGDTSGDYKKALLLLCGEDDGGGGSGGGGSGGGGSSEHKAKYENVTKDIFDLRDYYSGASKELKNVTGYRYSKGGKHYLIFDKNRKFTRVQIFGKDIERFKARKNPGLDIFVVKEAENRNGTVFSYGGVTKKNQDAYYDYINAPRFQIKRDEGDGIATYGRVHYIYKEEISLKELDFKLRQYLIQNFDLYKKFPKDSKIKVIMKDGGYYTFELNKKLQTNRMSDVIDGRNIEKIEANIR.

### 2.5. SSL5-AnxA5 and PSGL1 Binding Assay

The U937 concentration was adjusted to 1 × 10^6^ cells/mL. The cells were incubated with different concentrations of SSL5-AnxA5 fusion proteins to compete with the recombined P-selectin-Ig-PE (Thermo Fisher, Waltham, MA, USA, Cat No. 12-0626-82). After 20 min of incubation at room temperature, the amount of P-selectin-Ig-PE on the cell surface was measured by flow cytometry (Cytoflex, Beckman, Brea, CA, USA).

### 2.6. Flow Cytometry

In this study, except for apoptotic cells, which were detected using a kit, the following flow cytometry protocol was used for all the other flow cytometry tests. Add 50 μL of cell suspension to each tube. Mix the recommended amount of primary antibodies (each antibody’s concentration was recommended in the manufacturer’s instructions) in an appropriate volume of flow cytometry staining buffer and add to the cells, making the final staining volume 100 μL (50 μL cell sample + 50 μL antibody mixture). Gently vortex to mix. Incubate at 2–8 °C or on ice for at least 30 min, protected from light. Add flow cytometry staining buffer to wash the cells. For microtiter plates, use 2 mL/tube or 200 μL/well. Centrifuge at 400–600× *g* for 5 min at room temperature and discard the supernatant. Incubate at 2–8 °C or on ice for 60 min. Resuspend the cells in an appropriate volume of flow cytometry staining buffer. Dilute the appropriate amount of fluorochrome-conjugated secondary reagent in 100 μL of flow cytometry staining buffer and add to the cells. Incubate at 2–8 °C or on ice for at least 30 min, protected from light. Resuspend the cells in an appropriate volume of flow cytometry staining buffer. Resuspend the cells in an appropriate volume of flow cytometry staining buffer. Analyze the samples by flow cytometry (Cytoflex, Beckman).

### 2.7. Determining the Ability of SSL5-AnxA5 to Bind Apoptotic Cells by Flow Cytometry

To examine the ability of SSL5-AnxA5 to bind apoptotic cells, we used an AnxA5-FITC apoptosis detection kit according to the manufacturer’s protocol (Thermo Fisher, Cat No.V13242), adding the same dose of SSL5-AnxA5. We tested the double-fluorescent cell proportion among the apoptotic cells. The detail processes are as follows. Prepare 1X annexin-binding buffer. For example, for ∼10 assays, add 2 mL 5× annexin-binding buffer (Component C) to 8 mL deionized water. Prepare a 1 µM working solution of SYTOX™ Green stain, harvest the cells following apoptosis induction and wash in 1× annexin-binding buffer. Centrifuge the washed cells, discard the supernatant, and resuspend the cells at a concentration of ∼1 × 10^6^ cells/mL in 1× annexin-binding buffer. Add 5 µL R-PE annexin V (Component A) and 1 µL 1 µM SYTOX™ green stain working solution to each 100 µL of cell suspension. Incubate the cells at 37 °C in an atmosphere of 5% CO_2_ for 15 min. After the incubation period, add 400 µL of the 1× annexin-binding buffer, mix gently, and keep the samples on ice. And then the samples were tested by flow cytometry (Cytoflex, Beckman).

### 2.8. Endothelial Cell and Monocyte Cell-Binding Assay

P-selectin and phosphatidylserine in human umbilical vein endothelial cells (HUVECs) were induced to localize to the extracellular membrane, and a human myeloid leukemia mononuclear (THP-1) cell suspension labeled with BCECF fluorescent probe was used for co-incubation with activated HUVECs. After this interaction, the culture supernatant was discarded, the THP-1 cells combined with the HUVECs were retained, and their fluorescence values were measured. In addition, control PBS and different concentrations of SSL5-AnxA5 were added for co-incubation to observe its ability to inhibit the binding of the endothelial cells to monocytes and neutrophils.

### 2.9. Platelet–Leucocyte Interaction Assay

We collected fresh blood from healthy people (age > 18 years and no history of any diagnosed diseases) using TRAP for platelet activation. In addition, tagged platelets and a fluorescent secondary antibody against the white blood cells were added for incubation. Then, phosphate-buffered saline (PBS), the SSL5-AnxA5 fusion protein and a primary antibody against PSGL1 were added to the cells and incubated. We determined the percentage of cells showing double fluorescence among the monocytes and neutrophils. The ability of the fusion protein to prevent white blood cell adhesion to the platelets was tested. The CD45-FITC (Cat No. CD45-FITC) and CD42a-PE (Cat No. MA5-16698) flow cytometry antibodies were both purchased from eBioscience (San Diego, CA, USA).

### 2.10. Statistical Analysis

Statistical analyses were performed using SPSS statistics software for Windows version 19.0 (IBM SPSS Inc., Chicago, IL, USA). To compare unpaired groups, we used one-way ANOVA. The paired data were analyzed using a paired *t*-test. Dunnett’s test was used for multiple comparisons. The significance was set at *p* < 0.05.

## 3. Results

### 3.1. Identification of ACS and SAP Cell Clusters

Single-cell RNA sequencing was performed on cells extracted from the atherosclerotic plaques of patients with ACS and SAP. After quality control and data preprocessing, UMAP was used for dimensionality reduction and visualization, revealing 16 distinct clusters in both the ACS and SAP groups. The total number of cells analyzed was 2408, with 1537 cells from the ACS group and 871 cells from the SAP group (Figure 1A). Cell type annotation, using marker genes from the Human Primary Cell Atlas dataset, identified the major immune cell types, including CD4+ T cells, CD8+ T cells, monocytes, natural killer (NK) cells, B cells, macrophages, neutrophils, γδT cells, and platelets (Figure 1C). Batch effects were corrected using the Harmony algorithm, resulting in comparable cell type distributions between ACS and SAP, with no significant differences in the overall population distribution (Figure 1E). The convergence of the objective function during the clustering steps is illustrated in Appendix A, further demonstrating the effectiveness of the Harmony integration across datasets. However, detailed analysis of the cell type proportions revealed that, compared to SAP, the ACS group had higher proportions of monocytes, CD8+ T cells, B cells, and neutrophils, while the SAP group had higher proportions of CD4+ T cells and macrophages (Figure 1F). A heatmap displaying the marker gene expression across clusters (Figure 1B) further supported the annotation of the cell populations, confirming the distinct identities of each cluster. The expression levels of the representative marker genes for each major cell type were visualized in a dot plot (Figure 1D).

### 3.2. Differences in Intercellular Communication Between ACS and SAP Single-Cell Analysis

The bar plot shows the relative information flow of multiple signaling pathways between the ACS and SAP groups. Pathways such as *THBS*, *FN1*, *RESISTIN*, *COLLAGEN*, *BTLA*, *VCAM*, *ESAM*, *CD86*, *CD70*, *FASLG*, *IL1*, *CADM*, *MPZ*, *TIGIT*, *CD226*, *NECTIN*, *GAS*, and *CSF* are observed with different levels of signaling strength between ACS and SAP. Notably, the ACS group exhibits stronger signaling in pathways such as *VCAM*, *CD86*, and *FASLG*, while pathways like *COLLAGEN*, *FN1*, and *RESISTIN* show higher signaling in the SAP group. *SELPLG* and *ANNEXIN* show higher information flow in ACS compared to SAP (Figure 2A). The heatmaps present the overall and outgoing signaling patterns, respectively. In ACS, increased signaling activity is observed across multiple pathways, including *THBS*, *FN1*, *VCAM*, *CD86*, *CD226*, *FASLG*, *SELPLG*, and *ANNEXIN*, especially involving cell types such as monocytes, CD4+ T cells, and macrophages. These pathways display a higher signaling intensity in ACS compared to SAP. In SAP, pathways such as *COLLAGEN*, *FN1*, and *RESISTIN* show relatively higher signaling, particularly in outgoing signals from macrophages and neutrophils. The overall signaling patterns for *SELPLG* and *ANNEXIN* are more prominent in ACS, with *SELPLG* showing greater involvement of monocytes (Figure 2B,C). The circle plots depict the number of intercellular interactions between different cell types. ACS exhibits a greater number of interactions across several signaling pathways, suggesting more extensive cell–cell communication in ACS, especially between γδT cells, neutrophils, and platelets. SAP shows fewer intercellular interactions in these pathways (Figure 2D). The heatmaps complement this by comparing the differential number of interactions and the interaction strength. ACS shows a stronger interaction strength between γδT cells, neutrophils, and platelets, indicating that these cell types are more actively involved in intercellular communication in ACS. In contrast, SAP shows stronger interactions between monocytes, macrophages, CD8+ T cells, CD4+ T cells, and B cells, highlighting these cell types’ significant roles in cellular communication in SAP (Figure 2G). The *SELPLG* signaling pathway network demonstrates a higher number of interactions in ACS, particularly between monocytes, CD8+ T cells, and NK cells. The network is more limited in SAP, with fewer connections between these cell types (Figure 2E). The *ANNEXIN* signaling pathway network reveals stronger signaling from macrophages toward monocytes in ACS compared to SAP. This suggests a more active role of these cell types in transmitting *ANNEXIN*-mediated signals to monocytes in the ACS group. In SAP, the intensity of these signals is reduced, reflecting fewer interactions between these cell types (Figure 2F). Appendix A further illustrates the differential expression of critical components in the *SELPLG* and *ANNEXIN* signaling pathways across various cell types between the ACS and SAP groups. The heatmaps illustrate the differential expression of signaling components within the *SELPLG* pathway between ACS and SAP (Figure 2H). In ACS, higher expression levels of the *SELPLG* signaling components are observed in key cell types such as monocytes, CD8+ T cells, and NK cells, whereas the SAP group shows a reduced expression profile for these components, aligning with the overall reduced intercellular communication within this pathway.

### 3.3. The Preparation of the SSL5-AnxA5 Fusion Protein

To preserve the bioactivity of SSL5, AnxA5 was linked to the N-terminus of native SSL5 using a linker [(G4S)3] to separate the two domains (Figure 3A,B). The fusion protein cDNA sequence was inserted into pET-28a (+) to construct pET28a-SSL5-AnxA5-6xHis. The recombinant plasmids were identified by restriction endonuclease cleavage and DNA sequencing.

The recombinant proteins were expressed in BL21 (DE3) with the induction of 0.5 mmol/L IPTG. SSL5-AnxA5 was expressed in the soluble form. AnxA5 was purified by affinity chromatography. Figure 3C shows the target proteins after the purification of nickel agarose and the subsequent SDS-PAGE and Western blot analysis results reveal that the target proteins were expected to be recombinant proteins in our study (Figure 3D,E). Thus, fusion proteins consisting of SSL5 and AnxA5 domains were successfully prepared.

### 3.4. SSL5-AnxA5 Binds to PSGL1 and Inhibits Its Binding to P-Selectin

To determine whether SSL5-AnxA5 retains its ability to bind PSGL1, we performed a PSGL1 competition-based assay using the U937 cell line that constitutively expresses cell-surface PSGL1. The results showed that SSL5-AnxA5 can bind to U937 and inhibit P-selectin-Ig-PE binding to PSGL1 (saline vs. 0.25 μM group: 0 vs. 48.50 ± 5.49%, *p* = 0.009) (Figure 4A, B). This finding indicated that SSL5-AnxA5 may prevent lymphocytes from binding to inflammatory factors and partly block the conversion of some lymphocytes into inflammatory cells.

In the following part, we tested the binding capacity of SSL5-AnxA5 to apoptosis cells compared to AnxA5. By utilizing an AnxA5 apoptosis kit, we incubated the apoptosis-induced cells, AnxA5 fluorescence secondary antibody, fusion protein and His secondary antibody together. Microscopic observation and flow cytometry showed that the ratio of the double-fluorescent stained apoptosis protein was higher. Flow cytometry detected a much higher percentage of cells in the second and third quadrants and represented enhanced binding capability of fusion protein to apoptotic cells compared with AnxA5 (Figure 4C, D). All these data indicate that the fusion protein could bind to the serine expressed by the apoptotic cells and show the ability of SSL5-AnxA5 to bind apoptotic cells is superior to that of AnxA5.

### 3.5. SSL5-AnxA5 Prevented Endothelial Cells and Platelets from Combining with Inflammatory Cells

By utilizing the HUVECs and THP-1, we verified the inhibiting capacity of SSL5-AnxA5 in combining endothelial cells with inflammatory cells. The in vitro experiments showed that different concentrations of SSL5-AnxA5 inhibited the ability of HUVECs to bind to THP-1 and reduced the retention of THP-1 (SSL5-AnxA5 0.05 μM group vs. 0.5 μM group: 19.02 ± 11.37 vs. 39.22 ± 11.18, *p* = 0.006) (Figure 5A,B).

Then, we tested the ability of the fusion protein to prevent white blood cell adhesion to platelets and monocytes in healthy human blood. We used CD68 as a marker for human peripheral blood monocytes, CD45 as a marker for neutrophils, and CD42a as a marker for platelets, as analyzed via flow cytometry. Our results showed that the SSL-AnxA5 fusion protein partially prevented the adhesion of the monocytes and neutrophils to platelets (monocytes: saline vs. SSL5-AnxA5 vs. anti-PSGL1: 19.54 ± 2.03% vs. 12.39 ± 1.20% vs. 7.81 ± 0.36%, *p* = 0.006; neutrophils: saline vs. SSL5-AnxA5 vs. anti-PSGL1: 15.61 ± 0.60% vs. 8.27 ± 1.60% vs. 5.11 ± 0.68%, *p* = 0.002) (Figure 5C,D).

## 4. Discussion

In previous studies, annexin has been identified to be associated with tumors, the skeletal system and the immune system [22,23,24]. The *SELPLG* gene polymorphism is related to the PSGL1 level and influences the risk of atherosclerosis and coronary heart disease [25,26,27]. Ozaki et al. observed the up-regulation of the expression of PSGL1 in patients with ACS [28]. Our study verified the elevation of the *ANNEXIN* and *SELPLG* signaling pathway intensity in patients with SAP and ACS, paving the way for utilization of the fusion protein SSL5-AnxA5.

The P-selectin overexpression is a significant characteristic of vascular injury in CHD. A previous study has demonstrated that the integrating of P-selectin and PSGL1 was important in the occurrence and localization of pathological inflammation in an atherosclerosis mice model [29]. In a case control study, Shen et al. also identified that the high level of P-selectin was associated with the elevation of the coronary artery disease incidence [30]. The combination of overexpressed P-selectin and PSGL1 mediates a series of physiological process of neutrophils in inflammation, promoting the neutrophils to roll and aggregate on the surfaces of blood vessel through forming reversible bonds [31]. Meanwhile, the P-selectin also induced special signaling to take part in the activation of β2 integrins, contributing to the stable adhesion of neutrophils [32], which resulted in the stimulation of the inflammatory response and exacerbation from SAP to ACS. Therefore, inhibiting the interaction of PSGL1 and P-selectin may help reduce local inflammatory reactions. SSL5, as part of the fusion protein, was able to block this process [33]. By specifically binding to PSGL1, SSL5 can inhibit neutrophil rolling and extravasation and prevent endothelial cells from combining with inflammatory cells, alleviating the progression of CHD [34]. In our study, we innovatively constructed the SSL5-AnxA5 fusion protein to ensure the effective local drug concentration and we synchronously verified its efficacy in binding the PSGL1 and inhibiting the combination of endothelial cells with inflammatory cells, eliminating the biggest barrier in the treatment process for SSL5.

Endothelial cells and cardiomyocytes die during the progression of CHD and subsequently result in translocated phosphatidylserine. AnxA5 binds to translocated phosphatidylserine on the plasma membrane outer surface of cells undergoing apoptosis [35]. In our study, SSL5-AnxA5 allowed apoptosis-targeted aggregation by binding to the phosphatidylserine molecules at the outer surface of apoptotic cell membranes, and we also verified that the ability of the fusion protein to bind apoptotic endothelial cells is superior to that of AnxA5, which represented its outstanding efficiency and promising therapeutic prospects in CHD. More importantly, we verified the possibility that AnxA5 can be designed as a means of navigation to locate in apoptotic cell as part of the fusion protein, which broadens the horizons in drug transport pathways and future drug development related to an injured endothelium. Notably, the biofunction of the fusion protein may be influenced by the immunogenicity or the degradation during the process of transport. These are problems that cannot be overlooked in practical applications and more research is needed to explore them in the future.

Previous research indicated that SSL5 interacts with the platelet membrane GPIb receptor and activates platelets, which suggests that SSL5 first enhances platelet aggregation at the site of an injury [36]. However, other studies revealed that SSL5 decreased the interaction of platelets with neutrophils, abrogating the interaction required for enhancing thromboxane production in activated platelets [37]. In our study, platelet activation, platelet–monocyte aggregate formation and platelet–neutrophil aggregate formation were examined to examine the effect of SSL5-AnxA5 on platelet activation. The results showed that platelet–leukocyte aggregation was markedly reduced with SSL5-AnxA5 treatment. We speculate that AnxA5 reacts with phosphatidylserine in platelets and acts as an anticoagulant, which contributes to the third function of the fusion protein.

Our study is currently limited to in vitro cellular experiments, lacking in vivo validation of the SSL5-AnxA5 compound protein’s effects in animal models. In animal experiments, it is necessary to consider issues such as the immunogenicity, delivery efficiency, and protein degradation of the compound protein. Further improvements to the protein will be made to ultimately achieve precise and safe delivery of the protein to the disease site to exert its therapeutic effect. Another limitation of this study is that both the *SELPLG* and *ANNEXIN* signaling pathways were proposed through bioinformatics analysis and previous studies, and our study did not further experimentally explore the molecular mechanisms. Therefore, the molecular mechanisms through which the compound protein exerts its effects are inferred from the bioinformatics results and previous research findings. In the next phase of our research, in addition to conducting in vivo studies, we will also delve deeper into the molecular mechanisms underlying the action of this compound protein.

Taken together, the results of the present study provide a new strategy for CHD therapy, in which AnxA5 is used as a biologically active material to carry SSL5 directly to an injured artery. SSL5-AnxA5 is able to aggregate in injured arteries and inhibit leukocyte aggregation and adherence to endothelial cells. SSL5-AnxA5 may therefore be a promising approach for the treatment of CHD.

## 5. Conclusions

Our results demonstrated that the *ANNEXIN* and *SELPLG* signaling pathway intensity in patients with SAP and ACS, and SSL5-AnxA5, is able to prevent the interaction between apoptotic endothelial cells and platelets, as well as inflammatory cells. This study may provide a new strategy for delaying the progression of SAP patients to ACS.

## Figures and Tables

**Figure 1 biomedicines-13-00008-f001:**
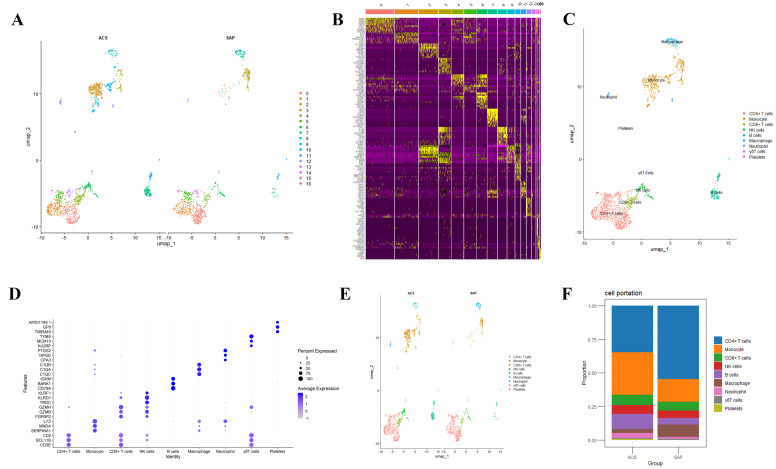
Analysis of single-cell RNA sequencing data in the ACS and SAP groups. (**A**) UMAP plots showing cell clusters from the ACS and SAP groups. Each dot represents a single cell, color-coded by cluster identity. Cells from both the ACS and SAP groups are distributed across multiple clusters, representing different cell types. (**B**) Heatmap displaying the expression levels of marker genes across the identified clusters (0–16). (**C**) UMAP plot showing the annotation of major cell types across both the ACS and SAP groups. (**D**) Dot plot showing the expression of key marker genes across the identified cell types. (**E**) Uniform manifold approximation and projection (UMAP) plot showing the distribution of major cell types in the ACS and SAP groups. (**F**) Stacked bar plot displaying the proportion of different cell types in the ACS and SAP groups. Each bar represents the relative abundance of major cell types, allowing for a comparison of the cell composition between the two groups.

**Figure 2 biomedicines-13-00008-f002:**
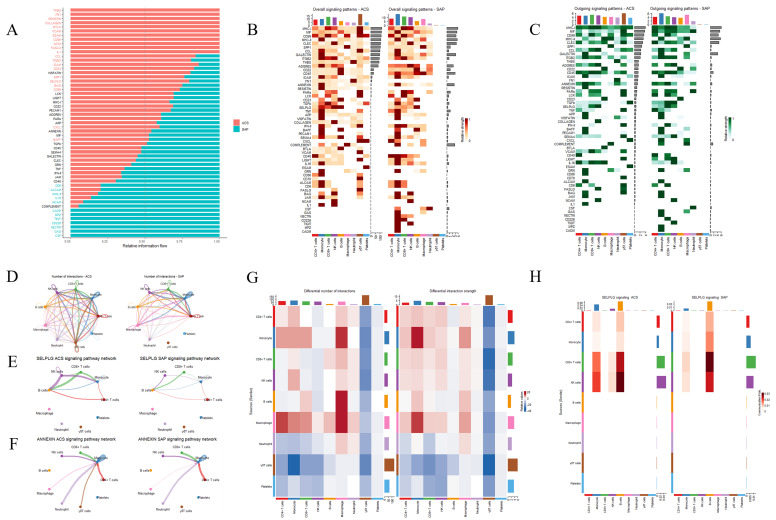
Comparative analysis of the signaling pathways between the ACS and SAP groups. (**A**) Bar plot comparing the relative information flow of the signaling pathways between the ACS (red) and SAP (blue) groups. Pathways are ranked based on their contribution to cell–cell communication. (**B**) Heatmap showing the overall signaling patterns for the ACS (**left**) and SAP (**right**) groups. Each square represents the communication strength between a source and a target cell type for a given pathway, with darker colors indicating stronger signaling interactions. (**C**) Heatmap of the outgoing signaling patterns, comparing the ACS (**left**) and SAP (**right**) groups. The signaling strength is color-coded, with darker green representing higher outgoing signaling levels for each cell type. (**D**) Circle plots showing the number of interactions between cell types in the ACS (**left**) and SAP (**right**) groups. The thickness of the lines represents the number of interactions, with thicker lines indicating stronger communication between cell types. (**E**) Network diagrams illustrating the *SELPLG* signaling pathway in the ACS (**left**) and SAP (**right**) groups. The lines represent interactions between different cell types, with the line thickness indicating the strength of the interaction. (**F**) Network diagrams illustrating the *ANNEXIN* signaling pathway in the ACS (**left**) and SAP (**right**) groups. (**G**) Heatmaps comparing the differential number of interactions (**left**) and the interaction strength (**right**) between cell types in the ACS and SAP groups. The color intensity represents the degree of difference, with red indicating higher interactions/strength in SAP and blue indicating higher values in ACS. (**H**) Heatmaps of the *SELPLG* signaling pathway showing the differential expression of signaling components between the ACS (**left**) and SAP (**right**) groups. The color intensity indicates the relative strength of the signaling, with red representing higher expression and blue representing lower expression.

**Figure 3 biomedicines-13-00008-f003:**
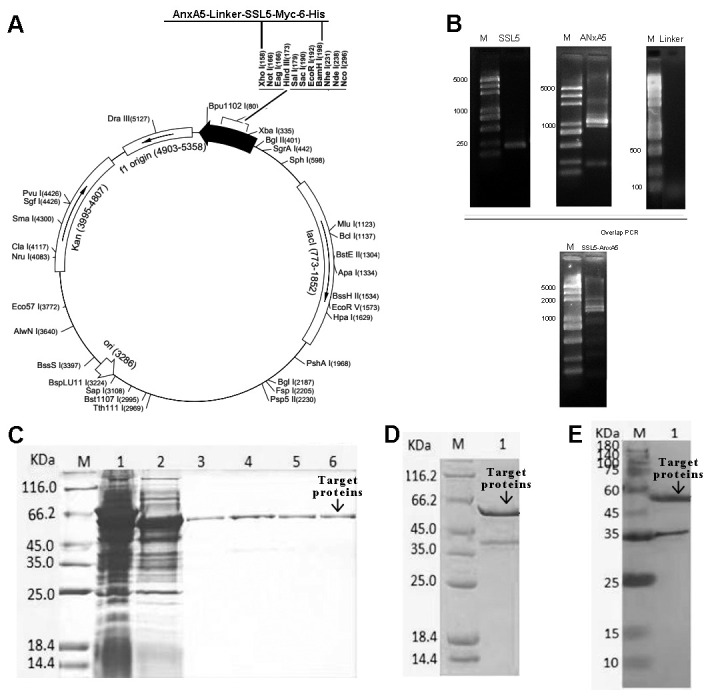
The preparation and purification of SSL5-AnxA5. (**A**) Schematic illustration of the SSL5-AnxA5 sequence. (**B**) Agarose gel electrophoresis analysis of SSL5, linker, AnxA5 and SSL5-AnxA5. (**C**) SDS-PAGE analysis of nickel agarose affinity chromatography purification of SSL5-AnxA5, M: protein marker; 1: loading sample; 2: flow-through; 3: 20 mM imidazole elution fraction; 4–5: 50 mM imidazole elution fractions; 6: 500 mM imidazole elution fraction. (**D**) SDS-PAGE analysis of the final protein purification, M: protein marker; 1: target proteins. (**E**) Western blot analysis of the final protein purification, M: protein marker; 1: target proteins.

**Figure 4 biomedicines-13-00008-f004:**
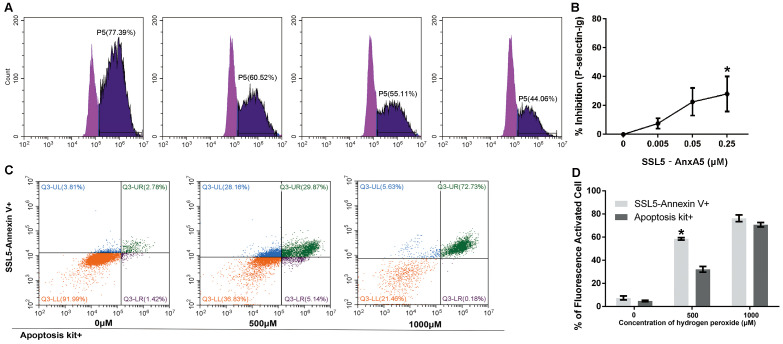
SSL5-AnxA5 binds to PSGL-1 and apoptosis cells. (**A**) The binding capacity of SSL5-AnxA5 to U937 cell-surface PSGL1. The dark purple area means the binding percentage of P-selectin with PSGL1 in the surface of U937. (**B**) Statistical results of the binding capacity of SSL5-AnxA5 to U937 cell-surface PSGL1 (n = 3, * *p* < 0.010; vs. saline group). (**C**) The binding capacity of SSL5-AnxA5 to apoptosis cells comparing to AnxA5. (**D**) Statistical results of the binding capacity of SSL5-AnxA5 to apoptosis cells comparing to AnxA5 (n = 3, * *p* < 0.010; vs. AnxA5 group).

**Figure 5 biomedicines-13-00008-f005:**
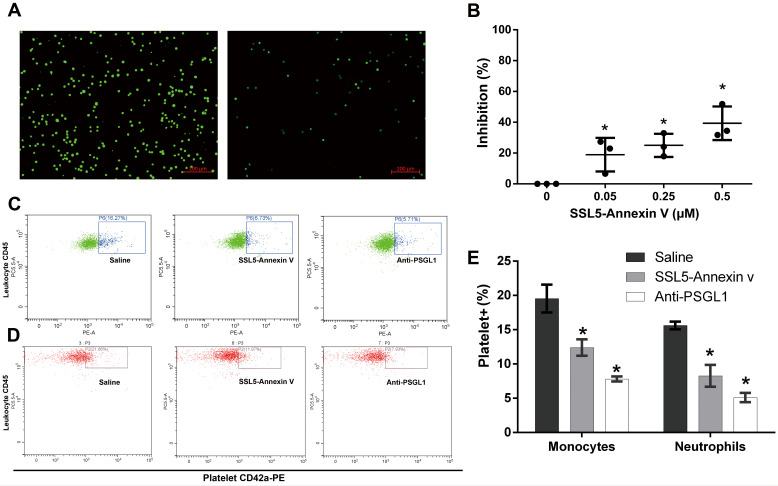
SSL5-AnxA5 inhibits the interaction of inflammatory cells. (**A**) SSL5-AnxA5 inhibits interaction of the HUVECs and THP-1. (**B**) Statistical results of the inhibition capacity of SSL5-AnxA5 in different concentrations (n = 3, * *p* < 0.010; vs. saline group). (**C**) SSL5-AnxA5 against the combination of activated platelets with neutrophils. (**D**) SSL5-AnxA5 against the combination of activated platelets with monocytes. (**E**) Statistical results of combination between monocytes or neutrophils with platelets in different groups (n = 3, * *p* < 0.010; vs. saline group).

## Data Availability

The data used to support the findings of this study are available from the corresponding author upon request.

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
