# Peer review of "SSL5-AnxA5 Fusion Protein Constructed Based on Human Atherosclerotic Plaque scRNA-Seq Data Preventing the Binding of Apoptotic Endothelial Cells, Platelets, and Inflammatory Cells"

_biomedicines, 2024, doi:10.3390/biomedicines13010008_

Round 1

Reviewer 1 Report

Comments and Suggestions for Authors

1. The introduction is detailed and effectively highlights the context of stable angina and acute coronary syndrome progression. However, some sentences require better structuring for clarity. It may benefit from including a specific hypothesis.

2. The methods section is comprehensive, but technical details about the preparation of the SSL5-AnxA5 fusion protein could use more explanation for reproducibility. Visual aids like flow diagrams for the methods might enhance reader comprehension.

3. The results are well-documented, with clear distinctions between ACS and SAP findings. However, the narrative occasionally lacks flow, making it difficult to follow the logical progression of experiments. Adding connecting statements between key findings may address this.

4. The discussion effectively links the results to the broader context of cardiovascular therapy. It could further emphasize the translational potential of the SSL5-AnxA5 fusion protein and its implications for therapeutic applications.

5. Figures and tables are informative but might benefit from more descriptive legends to facilitate standalone interpretation. Figures comparing binding efficacy across conditions could use consistent scaling for clarity.

6. The conclusion succinctly restates the study’s significance but does not sufficiently discuss limitations or future directions. Including these aspects could provide a balanced perspective.

7. References are extensive and relevant, though a few minor formatting inconsistencies were observed.

Reviewer 2 Report

Comments and Suggestions for Authors
  • Lacks concise explanation of why a fusion protein was chosen; more context on SSL5 and AnxA5 fusion needed.
  • Figures 1-5 need clearer captions and descriptions to explain their relevance; visual clarity of heatmaps and UMAP plots needs improvement.
  • Missing purification process details, like buffer composition and centrifugation parameters, limits reproducibility.
  • Flow cytometry experimental details insufficient; need specifics on replicates, parameters, and antibody concentrations.
  • Results on competition inhibition (Figure 4) and cell binding inhibition (Figure 5) lack adequate interpretation for clinical significance.
  • Unclear whether SSL5-AnxA5 effects were compared directly to non-treated or individual component controls.
  • Inconsistent terminology between 'PSGL1' and 'SELPLG,' causing confusion.
  • Discussion lacks consideration of limitations like immunogenicity, delivery efficiency, and protein degradation.
  • Mechanistic insights are speculative without supportive pathway analysis to back the inhibition effects.
  • Insufficient ethical transparency; lacks details on patient consent for human sample usage.
  • Mechanism of SSL5-AnxA5 preventing cell interactions not clearly elucidated; lacks downstream molecular analysis.
  • Conclusion overstates therapeutic potential without acknowledging limitations or preliminary nature of findings.
Comments on the Quality of English Language

it is ok 

Round 2

Reviewer 2 Report

Comments and Suggestions for Authors

Congratulation